



# A review of multivariate social vulnerability methodologies; a case study of the River Parrett catchment, Somerset

I. Willis[1]   and J. Fitton[2]

[1]Birkbeck, University of London, United Kingdom
5   [2]University of Glasgow, Glasgow, United Kingdom

*Correspondence to:* I. Willis (iain.willis@jbarisk.com)

**Abstract.** In the field of Disaster Risk Reduction (DRR), there exists a proliferation of research into different ways to measure, represent, and ultimately quantify a population's differential social vulnerability to natural hazards. Empirical decisions such as the choice of source data, variable selection, and weighting methodology can lead to large differences in the classification 10   and understanding of the 'at risk' population. This study demonstrates how three different quantitative methodologies (based on Cutter et al. (2003), Rygel et al. (2006), and Willis et al. (2010)) applied to the same England and Wales 2011 Census data variables in the geographical setting of the 2013/2014 floods of the Parrett river catchment, UK, lead to notable differences in vulnerability classification. Both the quantification of multivariate census data and resultant spatial patterns of vulnerability are shown to be highly sensitive to the weighting techniques employed in each method. The findings of such research highlight 15   the complexity of quantifying social vulnerability to natural hazards as well as the large uncertainty around communicating such findings to DRR practitioners.

## Introduction

The impacts of a natural hazard event upon a population vary considerably depending upon the socioeconomic attributes of the people exposed to the hazard i.e. social vulnerability (Yoon 2012; Zakour & Gillespie 2013). Social vulnerability can be 20   thought of as the degree to which a person is likely to be affected by a hazard, based upon their ability to prepare, cope, resist and recover from the hazard's impact (Twigg 2001; Cannon 1994). To support disaster risk reduction (DDR), it is important to quantify and spatially map a population's social vulnerability to natural hazards so that mitigation and adaptation strategies can target the most at risk populations (Rygel et al. 2006; Nelson et al. 2015; Yoon 2012).

There is a general consensus in social science about some of the main factors influencing an individual's social vulnerability 25   e.g. age, income, health, education level (Cutter 1996; Cutter et al. 2003; Adger et al. 2004; Wisner et al. 2004). Based on the concepts of political power, social capital, social networks and physical limitations, a distinction is made between the risk of natural perils and the antecedent conditions that may prevail and make some population groups more vulnerable than others. These notions were further outlined in the 'Hazards-of-Place' model by Cutter et al. (1996) to provide firstly, a conceptual understanding of how these influences interact, and then subsequently a quantitative methodology to identify and classify 30   social vulnerability. This later technique became a trademarked methodology, known as the Social Vulnerability Index (SoVI®). However, there has been no agreement on a set of social vulnerability indicators for environmental hazards to use within an index (Cutter et al. 2003; Yoon 2012). The data to include is constrained by the indicators relevance to the particular hazard(s) being assessed, and whether data are available and current (census data is often the primary data source). As a result, the number, and type of vulnerability indicators used within the construction of social vulnerability indices varies considerably 35   depending on the type of analysis, and methods used (Nelson et al. 2015).

Once the relevant vulnerability indicators have been selected to construct an index, they are combined into a single metric. However, Yoon (2012, p. 824) states that "*there is still no consensus…on the quantitative methodology best suited to assess social vulnerability*". Within the literature, the predominant method used is a multivariate factorial method, in the form of



principal component analysis (PCA) using census data e.g. Rygel et al. (2006), Boruff et al. (2005), Cutter et al. (2003), Clark
et al. (1998). Willis et al. (2010) use another method which utilised a commercial geodemographic (Experian Mosaic Italy)
classification as the main data source, and Gini coefficients to weight the vulnerability variables.

Yoon (2012) analysed the difference between a deductive and inductive approach when creating a vulnerability index, however
there has been no further research into comparing different vulnerability methodologies. Therefore, there is limited information
on whether all being equal, the different vulnerability methodologies classify the same people as highly vulnerable. The aim
of this paper is to compare the social vulnerability indices produced when using three published methodologies: a method
based on Cutter et al. (2003), a method using Pareto ranking based on Rygel et al. (2006), and a method with Gini coefficient
weighting based on Willis et al. (2010). The area of the Parrett river catchment, in Somerset, UK, which was severely flooded
in the winter of 2013/2014, will be used as a case study. If these approaches identify different populations as vulnerable, it
raises a number of questions about how the 'at risk' population is defined. This paper will firstly, review the chosen
vulnerability index methodologies, and describe the case study area. Secondly, the method used to compare the social
vulnerability indices will be detailed. Finally, the results will be presented, and discussed.

**Quantitative approaches to measure social vulnerability**

Cutter et al. (2003), Rygel et al. (2006), and Willis et al. (2010) utilise PCA but with different intent and application. PCA is
used to "*reduce the dimensionality of a data set consisting of a large number of interrelated variables, while retaining as much*
*as possible of the variation present in the data set*" (Jolliffe, 2002, p. 1). PCA is a useful tool when creating composite
vulnerability indices, as a number of vulnerability indicators are used, which are often correlated to various degrees. By using
PCA, it is intended that factors or components that inherently capture social vulnerability are created. Whilst Willis et al.
(2010) did not make explicit use of PCA extraction scores in their quantitative assessment of social vulnerability, multivariate
analysis was used in the screening and assessment of variables, hence its inclusion in this comparison.

Cutter et al. (2003) first used the SoVI approach to assess social vulnerability to a general environmental hazard using 1990
US census data, whereby 42 initial variables were reduced to 11 components using factor analysis (see Table 1 for further
information). On this basis, the 11 factors identified in PCA accounted for 76.4 % of the variance within the data. These
components were subsequently used to derive an overall social vulnerability index score (SoVI). The principle underlying the
methodology includes a binary assumption of the trend of specific vulnerability-related census variables. Variables included
in the initial assessment were assumed to have a positive or negative cardinality in their relationship to vulnerability. For
example, *non-white ethnicity* was considered to increase an individual's social vulnerability on the basis of historical studies
of disaster experience (Pulido 2000; Bolin 1993). Conversely, indicators relating to *wealth* are seen as negative factors,
reducing the relative social vulnerability score. Following this process of initial variable selection, PCA is then undertaken to
analyse the variables. The method used by Cutter et al. (2003) recommends the preservation of cardinality between vectors,
hence, any variables not correlated with the principal components of vulnerability are recommended to be removed and any
scores negatively correlated to vulnerability are inverted. Cutter et al. (2003) recommend a varimax orthogonal rotation is
undertaken to reduce the loading on the first component, as well as provide more independence among factors. Extraction
scores are then output for each factor in the data, and summed against the initial variables in an additive model to produce a
composite SoVI score.

Rygel et al. (2006) used a modified approach to the SoVI in their assessment of areas vulnerable to hurricane storm surge
(Table 1). Following PCA and subsequent varimax rotation of the variables, it is proposed that Pareto ranking is applied to the
PCA extraction scores (see Rygel et al. (2006) for a fuller explanation of the theory of Pareto ranking). The basis of applying
a Pareto distribution across the vulnerability scores is to remove the requirement of individually weighted scores, and thus,


overcome concerns about systematic bias. Each component score is then ranked on the basis of a user defined interval (19 in the original method) and an overall ranking is determined.

**Table 1: Summary of the three social vulnerability methods applied within this paper.**

|  | Cutter et al. (2003) | Rygel et al. (2006) | Willis et al. (2010) |
|---|---|---|---|
| **Hazard** | General Environmental Hazards | Hurricane Storm Surges | Volcanic Eruption |
| **Study Area** | United States | The Hampton Roads, Virginia, United States | Mount Vesuvius, Naples, Italy |
| **Data Source** | 1990 United States Census | 2000 United States Census | Experian Mosaic Italy |
| **Spatial Unit** | County | Census Unit | Census Unit |
| **Number of Indictors** | 42 | 57 | 7 |
| **Indicator Format** | Percentages, Per Capita, Density Functions | Percentages, Areal Densities | Propensity index score |
| **PCA Factors** | 11 – explained 76.4 % of variance (used Varimax rotation) | 3 – explained 50.83 % of variance (used Varimax orthogonal) | Did not directly use PCA |
| **Vulnerability Dimensions** | Personal Wealth, Age, Density of the Built Environment, Single-Sector economic dependence, Housing stock and tenancy, Race (African American, Asian) Ethnicity (Hispanic, Native American, Occupation, Infrastructure dependence | Poverty, Immigrants, Old Age/Disabilities | Evacuation, Financial Recovery, Building vulnerability, Access to resources |
| **Method Used to Combine Indicators** | Addition of Extraction Scores | Pareto Ranking of Factor Scores | Addition and Averaging of Weighted (using Gini Coefficients) Index Score |

Willis et al. (2010) analysed Italian census areas around Mount Vesuvius using Mosaic Italy 2007 geodemographic index scores (Table 1). Instead of using PCA extraction scores, it was proposed that an additive model was applied, whereby social

vulnerability variables were weighted according to their economic Gini coefficient value to provide a composite score. The concept of this approach being that the Gini coefficient provides a precise measure of variable discrimination, and therefore an appropriate weighting tool to assign some vulnerability variables with higher loadings than others.

**The River Parrett Catchment**

For the purposes of comparing the alternative methodologies, it was decided that a relevant geographical setting be used to

apply the vulnerability scores within a pertinent historic context. By doing so, it was proposed that meaningful assessment could be undertaken of the results within a realistic natural hazard setting. Given the low lying nature of the area and it's prevalence to flood risk, the Parrett catchment, Somerset, UK was chosen as the case study area for this research (Figure 1).

The UK experienced an unprecedented level of rainfall during the winter of 2013/2014, resulting in the flooding of 65 km$^2$ of the Somerset Levels area of the River Parrett catchment area (Environment Agency 2015). Approximately 600 properties were

flooded during this period, leaving a number of towns and villages cut off due to the high floodwaters. Flood waters persisted until March 2014 and the damage witnessed raised a national debate about the lack of dredging in the rivers throughout the Parrett catchment (Coghlan, 2014; Envionment Agency 2015;). This political pressure resulted in ministerial intervention and the subsequent production of 'The Somerset Levels and Moors Flood Action Plan', a 20-year scheme to mitigate future flood potential and increase the level of funding for flood management in the region (Somerset County Council 2014). Aside from





the flooding witnessed in 2013/2014, previous reports have estimated that 3,300 properties are potentially exposed to a 1% annual probability flood event within the catchment, with this possibly rising to over 6,600 properties in the future, due to the impacts of climate change (Environment Agency 2009). There is evidence that this rise is likely to occur as the flooding in England and Wales in 2013/2014 shown to have been linked to human-induced climate change (Schaller et al. 2016).

Alongside the physical damage of the Somerset Levels flooding, there has been limited consideration of the social vulnerability

of those communities affected. Within the River Parrett catchment area there are a range of socioeconomic profiles, and while many of the most deprived communities (those located in urbanised areas such as Yeovil, Taunton and Bridgewater) were not adversely impacted, flood risk potential remains high. In the wider context of flood risk management, England and Knox (2015, p, 7) show that in England *"levels of planned expenditure in flood risk management to 2021 do not appear to align with areas of significant flood disadvantage, or with wider deprivation",* i.e. the social vulnerability of the population potentially

impacted by flooding currently has no bearing on spending decisions. In this instance, vulnerability to flooding used by England and Knox (2015) was derived using a method based on Cutter el al. (2003) by Lindley et al. (2011).

To help confine the research to the flood risk case study area, a GIS spatial extent, as seen in Figure 1, was delineated for the River Parrett catchment area and used as the bounding area to select the England and Wales Census Output Areas within the catchment. Similarly, a flood footprint relating to the 2013/2014 event was digitised as a GIS layer based on the maximum

extent identified by the Environment Agency (2014). This extent provided the basis of comparison results highlighted in Figure 7 and Table 6.

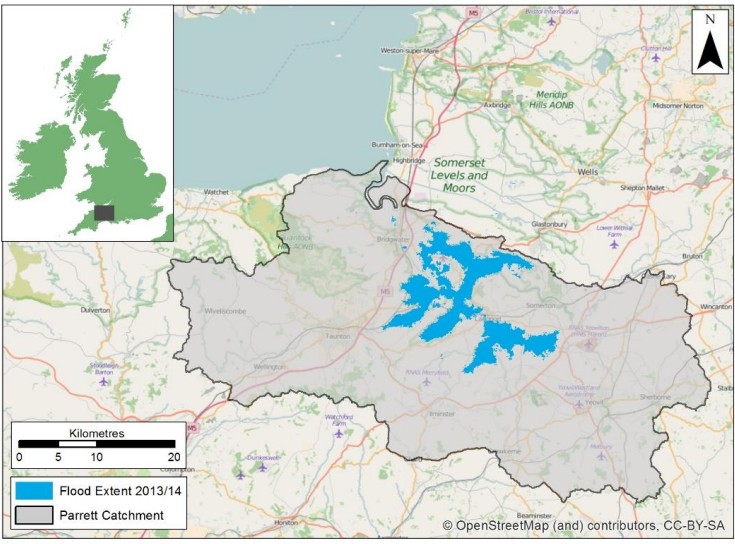

**Figure 1: The location of the Parrett catchment, within the Somerset Levels area of south-west UK. The extent of the flooding flood in 2013/2014 is also shown. Backdrop mapping provided by OpenStreetMap.**

**A standardised methodology to compare quantitative approaches**

The principle aim of this study was to devise a methodology that could allow the different quantitative social vulnerability methods (outlined earlier in this paper) to be compared in a consistent manner. For this purpose, it was necessary to devise a repeatable process, whereby only the weighting of the variables would be changed to recognise each different methodology.



**Selection of vulnerability indicators**

Data for this study were taken from the *2011 Area Classification for Output Areas*, a joint venture between the Office of National Statistics (ONS) and University College London (UCL) to help disseminate and inform researchers about the 2011 Output Area Classification (OAC2011). The OAC2011 is a neighbourhood classification based on the most recent UK census, conducted in March 2011. This study has made use of the UK Output Area spatial boundaries (in ESRI shapefile format) as well as census variable data (at Output Area level) used to construct the OAC2011 neighbourhood classification available from http://geogale.github.io/2011OAC/.

The England and Wales census data was used in this study which comprises of 232,296 Output Areas (ONS 2011). It's important to note that not all data collected from the census is used in the creation of the OAC2011. To devise the neighbourhood classification, a process of variable selection was used to help determine data inter-dependencies, correlations, and other factors that may affect the clustering process (Vickers et al. 2005). Of the 59 census variables (including derived statistics) used to create the OAC2011, it was determined that only seven specific data variables would be suitable for inclusion in the social vulnerability classification comparison (Table 2).

**Table 2: 2011 UK Census data variables used as the indicators to assess social vulnerability to flooding**

| Census Code | Indicator Description | +/- Effect on Social Vulnerability | Supporting Literature |
|---|---|---|---|
| k001 | Persons aged 0 to 4 | Negative | McMaster and Johnson 1987; Lew & Wetli 1996 |
| k005 | Persons aged 65 to 89 | Negative | McMaster and Johnson 1987; Lew & Wetli 1996 |
| k007 | Number of persons per hectare | Negative | Johnson and Ziegler 1986; Chakraborty et al. 2005; Dow and Cutter 2002 |
| k023 | Main language is not English and cannot speak English well or at all | Negative | Pulido 2000; Elliot and Pais 2006 |
| k033 | Households who are social renting | Negative | Burton et al. 1992 |
| k035 | Individuals day-to-day activities limited a lot or a little (Standardised Illness Ratio) | Negative | Morrow 1999; Dwyer et al. 2004 |
| k045 | Persons aged between 16 and 74 who are unemployed | Negative | Burton et al. 1992 |

There were two main reasons for the seven initial indicators shown in Table 2. Firstly, as the focus of the study was to determine the difference that alternative weighting mechanisms may have on vulnerability scores, using fewer indicators made it easier to infer the influence of each methodology being reviewed. Secondly, not all census variables were eligible for inclusion in this study given that the focus was on determining factors that impact a neighbourhood's social vulnerability during extreme flooding. Whilst not exhaustive, Table 2 also provides example studies of where age, ethnicity, and disability have been shown to impact social vulnerability to support the selection of indicators within this study. Table 3 shows the correlation between the selected vulnerability indictors, with 'Persons aged 65 to 89' and 'Individuals day-to-day activities limited a lot or a little' (k005 and k035) showing the strongest relationship (0.687). Table 3 demonstrates that none of the variables show particularly high degrees of correlation, and therefore none of the indicators were removed from the analysis on this basis.





**Table 3: Correlation between input vulnerability indicators**

|      | k001   | k005   | k007   | k023   | k033   | k035  |
|------|--------|--------|--------|--------|--------|-------|
| **k005** | -0.501 |        |        |        |        |       |
| **k007** | 0.282  | -0.62  |        |        |        |       |
| **k023** | 0.644  | -0.518 | 0.617  |        |        |       |
| **k033** | -0.225 | -0.565 | 0.599  | 0.201  |        |       |
| **k035** | -0.044 | 0.687  | -0.162 | -0.133 | -0.499 |       |
| **k045** | 0.685  | -0.364 | 0.591  | 0.586  | -0.027 | 0.389 |

**Data standardisation**

The data from the England and Wales census are not in a standardised format or description. For example, age group data (K001 and K005) were initially provided as numerical counts within the Output Area. These values had to then be converted to a percentage with respect to the overall population recorded within a given Output Area. Alternatively, population density (K007) was recorded as a measure of people per hectare, and disability (K045) noted according to the standardised illness ratio (SIR). Whilst these data formats are relevant for their respective measures of a phenomenon, they would not have been suitable for multivariate analysis, correlation tests or weighting variables against one another. For this purpose, it was necessary to firstly standardise the data into a homogenous format. There are commonly two methods employed to standardise data, including Z-scores or Range standardisation (Wallace and Denham 1996). In this case, the Range standardisation method was applied as it was also used in the construction of the OAC2011, and was therefore determined to be the most relevant to this research (Vickers et al. 2006). The Range standardisation is shown in equation (1), whereby the standardised observation ($xn$) is calculated as a ratio from the maximum and minimum observations for a given variable. This leads to all observation values being classified between 0-1.

$$xn = \frac{x - xmin}{xmax - xmin} \qquad\qquad (1)$$


**Exploratory Principal Component Analysis**

To help assess the cardinality of the data variables as well as their inter-dependency and variance, PCA was undertaken on the standardised census data. An initial PCA showed that three components accounted for 91% of the overall variance in the data, with the first component accounting for 48%. Further analysis of this component showed that the variables *population density* (K007), *non-English speaking* (K023) and *unemployment* (K045) were highly correlated and had the largest component loadings. Conversely, the variables *age 65-89* (K005) and *Standardised Illness Ratio* (K035) showed negative loadings for the same component. This pattern of correlation among variables can be seen further in Figure 2 whereby the cardinality of vectors are positively aligned for K007, K023, K001, and K045. Conversely, K005 showed strong negative correlation with all variables apart from K035.





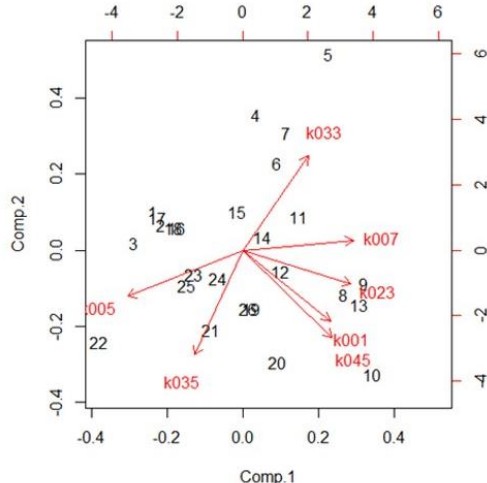


**Figure 2: Left – correlation plots of the UK census variables. Right – biplot of the component vectors**

**Assess cardinality of vectors**

The method used by Cutter et al. (2003) proposed that following analysis, only vectors with the same cardinality should be retained for inclusion in the vulnerability index. This is based around the concept that each of the variables remaining is

correlated with vulnerability and therefore, an index can be produced by summing these variables with the component score. It should be noted that Cutter's approach states that where a variable is understood to reduce vulnerability due to having a positive effect (such as a household's wealth/income), the variable should be inverted to become a negative score.

Although Rygel et al. (2006) and Willis et al. (2010) did not espouse reducing variables on the basis of PCA cardinality, it was necessary to remove variables K005 and K033 from further inclusion to ensure a consistent methodology was maintained. As

the comparison methodologies outlined in Cutter et al. (2003) and Rygel et al. (2006) made use of rotated component scores as an input to the vulnerability assessment, a similar step would be required in this research to maintain continuity of the methods being compared. In accordance with the prescriptive methodologies outlined in these applications of multivariate analysis, the remaining five variables were subsequently rotated using a varimax rotation, and the component scores extracted for each Output Area. The extracted score became a new input variable (referred to hereafter as *PCA Vulnerability Score*) and

was used in the creation of the vulnerability indices outlined in the results section.

**Gini Coefficients**

Figure 3 provides a summary of the Lorenz curves for each of the variables. Lorenz curves provide a graphical illustration of the Gini Coefficient and thus show the cumulative distribution of a variable within a population (Gastwirth 1972). The greater the area between the curve and the 'line of equality' represents how skewed or discriminatory a variable is within a given

population.




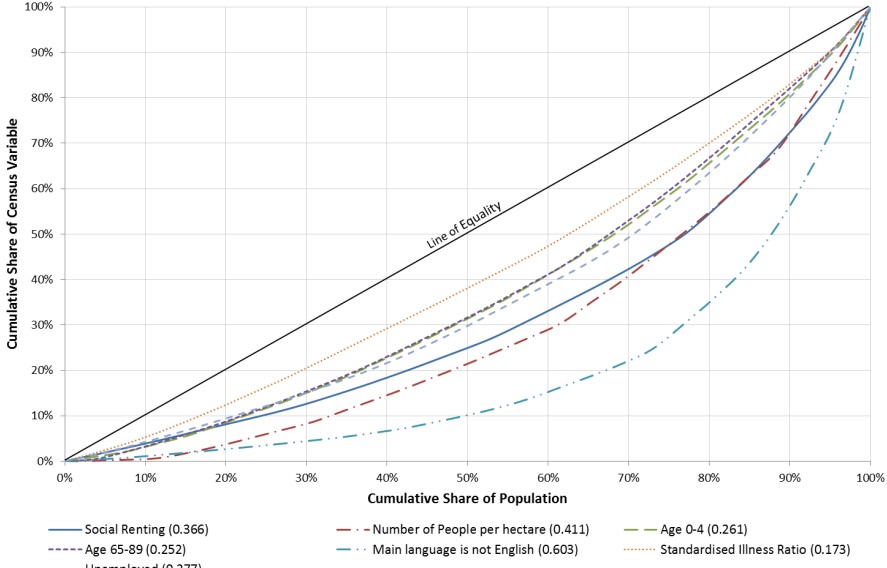

**Figure 3: Lorenz curves of Output Area Classification (OAC) selected to assess social vulnerability to flooding. Gini coefficient is shown within the graph legend.**

Figure 3 highlights how UK Census variables, such as *Main language is not English* (K023) are disproportionately distributed among the OAC classification groups. In comparison, *Standardised Illness Ratio* (K035) is much less skewed among these profiles. This was further highlighted by the corresponding Gini coefficient values, 0.603 and 0.173 respectively for the variables. This was calculated using a generalised method (Bellù and Liberati 2006) whereby values closer to 1 represent greater inequality than values closer to 0.

**Apply weighting**

Though the alternative methodologies shared many similarities, they also had distinct differences in their selection, weighting and summation of the input variables. The application of each of methodology to the standardised census data is summarised in Table 4 below:

Table 4: Summary of how the social vulnerability index is constructed using the three different methods

|  | Cutter et al. (2003) | Rygel et al. (2006) | Willis et al. (2010) |
|---|---|---|---|
| **Variables** | K001+K007+K023+K045+*PCA Vulnerability Score* | *PCA Vulnerability Score* | K001+K007+K023+K045 |
| **Process** | Additive | Pareto ranking (*100 intervals*) | Additive |
| **Output** | Index ($X_i$ / $X_{mean}$ *100) | Index ($X_i$ / $X_{mean}$ *100) | Index ($X_i$ / $X_{mean}$ *100) |

In terms of input variables, Cutter's *Social Vulnerability Recipe* recommends an additive approach, whereby the individual census variables are added together along with the PCA extraction score created during rotation of the variables (Cutter 2008). Willis et al. (2010) have a similar approach in summing variables but do not use the additional extraction scores. Conversely, Rygel et al. (2006) do not use any of the input census variables and instead use only the vulnerability extraction score to provide a summary of the Output Area. Rygel et al. (2006) recommend applying a Pareto ranking to the extraction scores, which involves placing observations into discrete 'blocks' or ranges. Depending on how many components are input, the data can be ranked on multiple variables. The final step in the process is to sum the ranks and provide an overall weighting. The



intention of doing this is to reduce the skew effect that one variable may have on the overall result. The procedure of Pareto ranking is highly subjective in the choice of how many ranks or intervals are created for the given distribution of observations. Based on the proportion of intervals that Rygel et al. (2006) used in their study of US counties, it was decided that 100 intervals

would provide an approximate correlation for the Output Areas based on the PCA Vulnerability Score.

The final methodological step was to provide a normalised output from each technique to compare the results in a systematic manner. For this purpose, a propensity index was used. A propensity index is commonly used in geodemographics to convey relative variable scores and reduce any apparent bias between variable distributions. Equation (2) below summarises how the index score for a variable ($xi$) is calculated from a ratio of the observation value ($x$) from the variable mean average ($\bar{x}$)

multiplied by 100.

$$xi = \frac{x}{\bar{x}} \times 100 \hspace{6cm} (2)$$



## Results

**Distribution of social vulnerability scores**

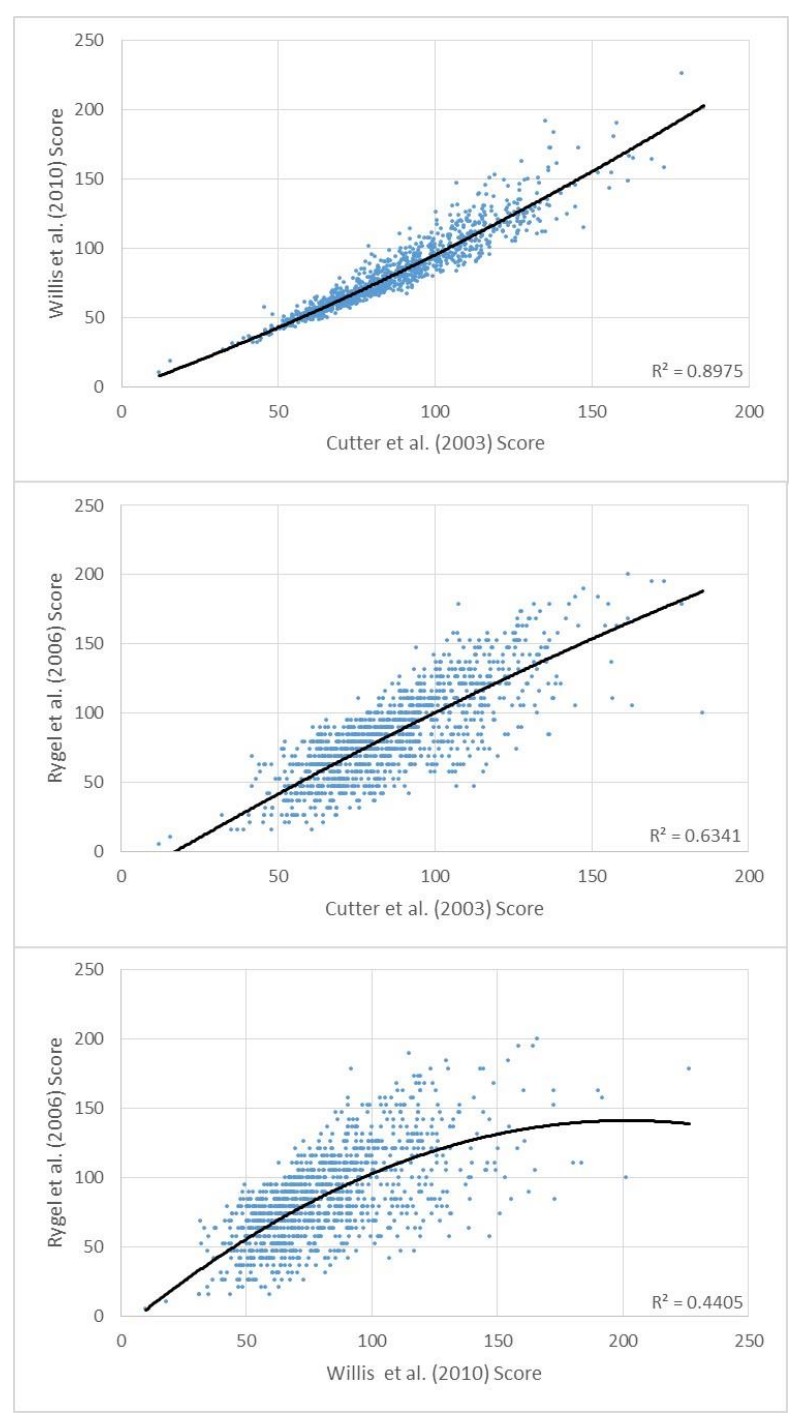

**Figure 4: Correlation of social vulnerability index scores for the Parrett Catchment. Trend lines are polynomial.**



Figure 4 shows the correlation between the social vulnerability index scores derived from each of the three methods. The social vulnerability scores from Cutter et al. (2003) and Willis et al. (2010) show relationship close to linear with a strong correlation evident ($R^2 = 0.8975$). Comparison of the Cutter et al. (2003) and Rygel et al. (2006) scores again show an almost linear relationship but the data show less correlation ($R^2 = 0.6341$). The relationship between the Willis et al. (2010) and Rygel et

al. (2006) results show a much weaker correlation ($R^2 = 0.4405$). The Willis et al. (2010) scores show that the method produces a more extreme classification of scores, than the Rygel et al. (2006) scores, shown by the flattening of the trend line. Figure 5 highlights the distribution of vulnerability scores across the Output Areas for all methodologies for the Parrett catchment. Whilst the graph shows a correlation between the Gini coefficient approach (Willis et al. 2010) and Cutter's method (Cutter et al. 2003), Rygel's Pareto ranking method (2006) displays a greater variation in the classification of the same Output Areas; the

choice of 100 rank intervals used in the method appears paramount to the relative distribution of these scores. This point is further shown in the correlation plots of Figure 4 by the 'stepped' pattern of the Rygel et al. (2006) data and in Table 5 with the standard deviation for the Rygel et al. (2006) approach being 33.1, in comparison to the Willis et al. (2010) method (27.3), and Cutter et al. (2003) approach (23.3) for the Parrett catchment. Interestingly, this relationship is not the same when considering all of the England and Wales Output Areas, whereby the Willis et al. (2010) method resulted in the highest standard

deviation (42.6). This last point appears due to the loading factor the Willis et al. (2010) method had on vulnerability scores that are greater than 100, thus leading to outlier scores. The Cutter et al. (2003) method showed the lowest standard deviation at all spatial scales along with the highest mean score (87.5) of vulnerability in the Parrett catchment, when compared to the other techniques.

**Table 5: Comparison of mean and standard deviations of the social vulnerability index scores by OAC 2011 classification within the**
**Parrett catchment. The mean and standard deviation of the England and Wales (E & W) is shown for comparison.**

| | OAC2011 Supergroup Classification | Number of Output Areas | Cutter et al. (2003) Mean Score | Willis et al. (2010) Mean Score | Rygel et al. (2006) Mean Score |
|---|---|---|---|---|---|
| **Parrett Catchment** | Constrained City Dwellers | 78 | 124.8 | 123.6 | 138.3 |
| | Cosmopolitans | 12 | 87.3 | 94.5 | 95.6 |
| | Hard-Pressed Living | 258 | 102.4 | 96.9 | 105.4 |
| | Multicultural Metropolitans | 5 | 125.6 | 148.2 | 102.1 |
| | Rural Residents | 388 | 72.5 | 63.7 | 70.0 |
| | Suburbanites | 154 | 73.8 | 66.5 | 68.2 |
| | Urbanites | 223 | 91.8 | 91.5 | 82.4 |
| | Total | 1,118 | 87.5 | 82.2 | 85.6 |
| | Standard Deviation | - | 23.3 | 27.3 | 33.1 |
| **E & W** | Total | 232,296 | 100 | 100 | 100 |
| | Standard Deviation | - | 32.5 | 42.6 | 42.3 |

In terms of the spatial distribution of scores, the three comparative methodologies show a high degree of correlation with regard to their urban-rural pattern of vulnerability scoring (Table 5). Vulnerability index scores greater than 100 were largely constrained to the centres of greatest population density, most notably the large Somerset towns of Taunton, Bridgwater, and

Yeovil. Table 5 shows that the highest average social vulnerability scores across the three methods are found in output areas classed by the OAC2011 classification as 'Constrained City Dwellers' and 'Multicultural Metropolitans'. Similarly, and despite subtle differences in the magnitude of scoring, spatial correlation was noted to be closer between Cutter et al. (2003) and Willis et al. (2010) in comparison to Rygel et al. (2006).




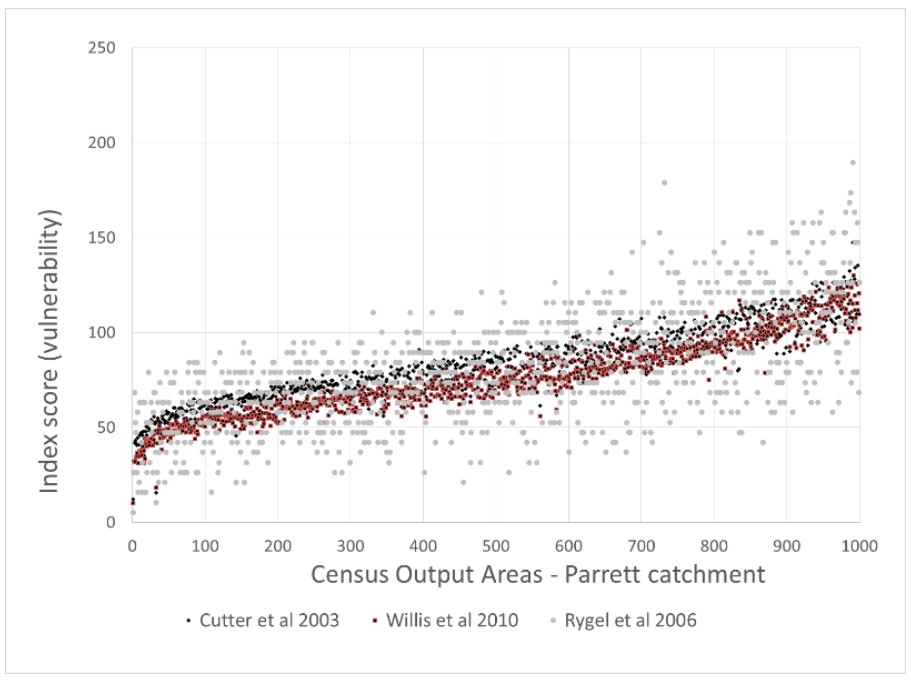

**Figure 5: Output Area comparison of social vulnerability index scores for the Parrett catchment**

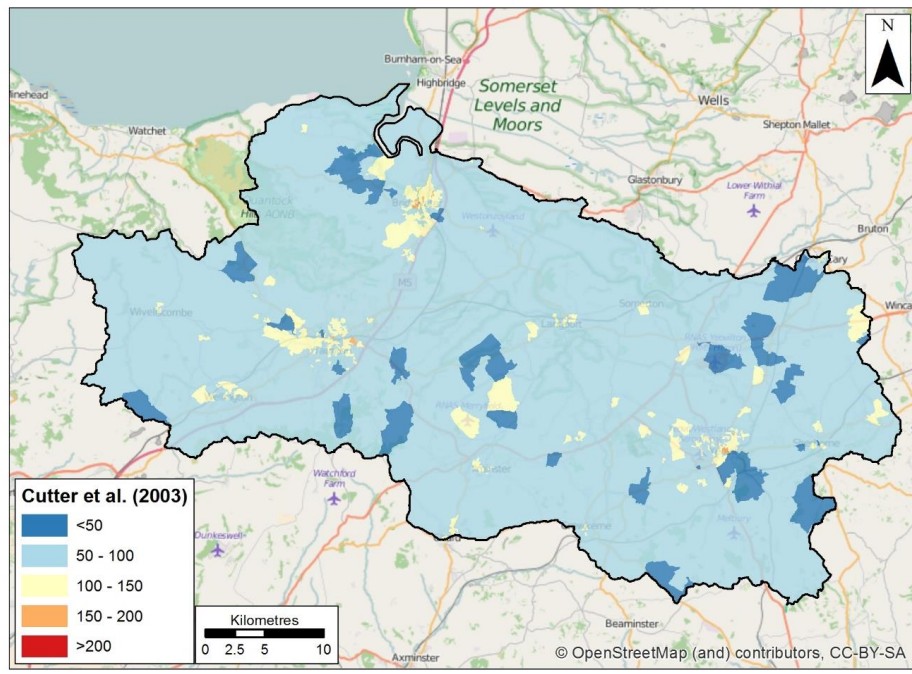

**Figure 6a: Spatial analysis of social vulnerability index based on the Cutter et al. (2003) methodology for the Parrett catchment, UK**





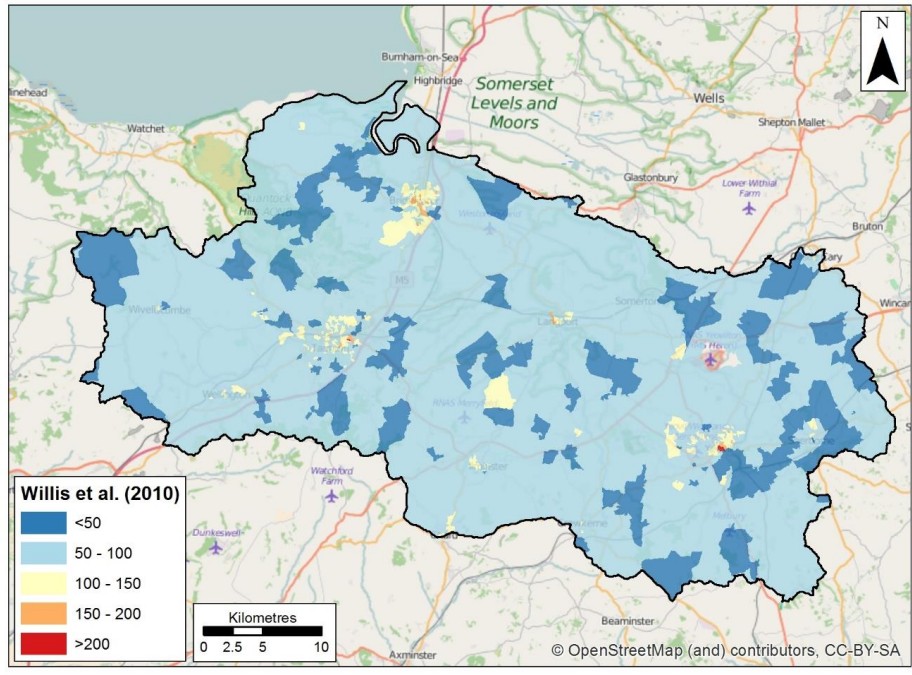


**Figure 6b: Spatial analysis of social vulnerability index based on the Willis et al. (2010) methodology for the Parrett catchment, UK**

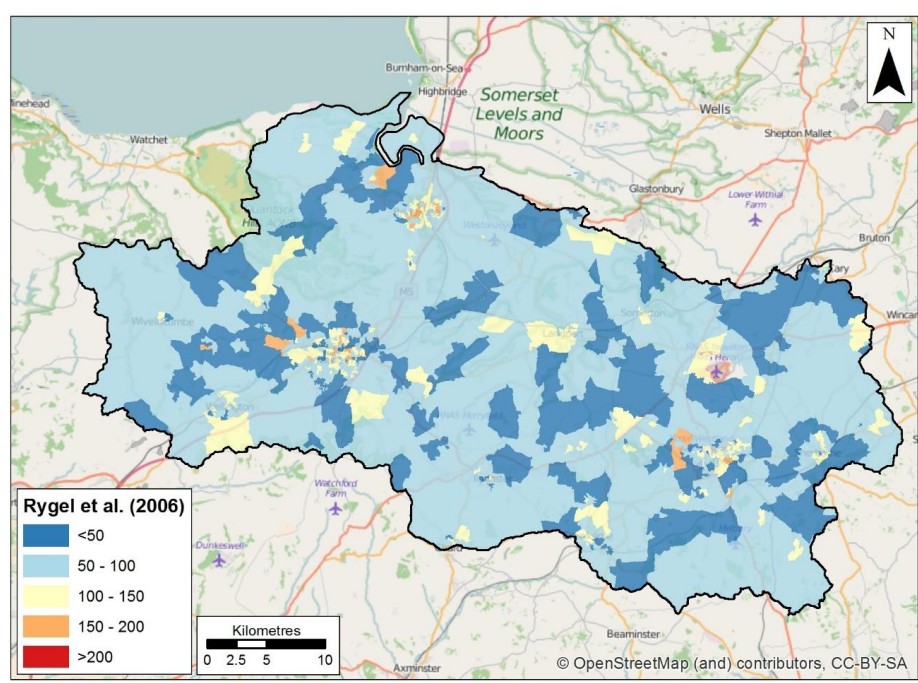

**Figure 6c: Spatial analysis of social vulnerability index based on the Rygel et al. (2006) methodology for the Parrett catchment, UK**





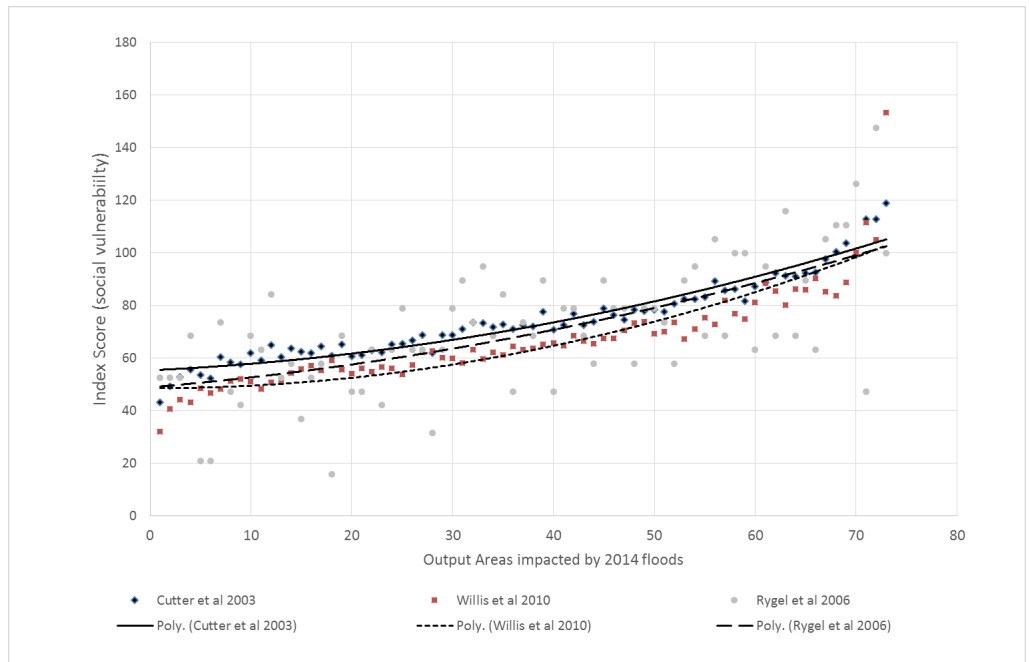

**Figure 6: Output Area comparison of social vulnerability index scores for the areas impacted by the 2013/2014 flooding**

**Table 6: Analysis of the areas impacted by the 2013/2014 flooding of the Somerset Levels.**

| OAC2011 Supergroup Classification | Number of Output Areas | Cutter et al. (2003) Mean Score | Willis et al. (2010) Mean Score | Rygel et al. (2006) Mean Score |
|---|---|---|---|---|
| Hard-Pressed Living | 2 | 102.1 | 86.3 | 110.5 |
| Rural Residents | 67 | 71.9 | 63.9 | 67.0 |
| Suburbanites | 1 | 81.6 | 74.9 | 100.0 |
| Urbanites | 3 | 110.3 | 119.5 | 124.6 |
| Total | 73 | 74.5 | 66.9 | 71.0 |
| Standard Deviation | - | 15.4 | 18.2 | 24.1 |

The distribution of social vulnerability in the Parrett catchment is repeated at the smaller scale when an assessment of the output areas that experienced flooding in 2013/2014 flood are considered (flood extent is shown in Figure 1). The flooding impacted upon a total of 73 output areas with the majority (67) of these output areas categorised as 'Rural Residents' according to the OAC2011 Supergroup classification (Table 6). The average social vulnerability score across the three methods within the Rural Residents classification is 67.6, considerably below the England and Wales mean score of 100. This assessment

demonstrates that the people impacted by the flooding in 2013/2014 would most likely be considered to be less vulnerable than the majority of the England and Wales population. Using a smaller spatial scale to compare the three methods shows that a relatively consistent interpretation about the social vulnerability can be derived. However, as with the Parrett catchment analysis, the Rygel et al. (2006) method has a higher standard deviation than the two other methods. This is supported by Figure 6 which shows that the social vulnerability score derived from the Rygel et al. (2006) method of individual output areas

is extremely erratic, with the Cutter et al. (2003) and Willis et al. (2010) showing a more consistent relationship.




**Conclusion**

The three methods presented within the study are consistent when considering the mean scores and interpreting the general picture of social vulnerability within a geographic area. However, at the level of census output area level, the method based on the Rygel et al. (2006) method produces a social vulnerability classification that differs markedly from the results of the Cutter et al. (2003), and Willis et al. (2010). This research demonstrates the complexity in quantitatively defining the 'at risk' population in terms of social vulnerability to natural hazards. Despite applying alternative methodologies to standardised variable data in a confined geographical setting, differences in the classification and interpretation of the most vulnerable are shown to be evident. The study showed that the application and subsequent decision-making on the basis of Principle Component Analysis (PCA) results can lead to the creation of very different, but equally plausible methodologies to define vulnerable populations in the same study area. The subjective choices of whether to apply Pareto ranks, PCA rotation, and summation methods are just small examples of the relative impact such decisions may have on both the locality and quantitative value associated with risk. For example, Pareto ranking used within the Rygel et al. (2006) method was shown to lead to greater heterogeneity of scores, but arguably, less precision in the quantification of risk. The application of a Gini coefficient use by Willis et al. (2010) may lead to outliers through the exponential loading of higher vulnerability scores, however, the concept of an inclusive methodology could be argued to be equally as relevant as the non-selection approach based on the PCA cardinality.

Whilst recognising the uncertainty that various statistical methods impose on such classifications, it is also important to be mindful that the fundamental qualitative assumptions underlining social vulnerability are perhaps the first source of uncertainty in this process. For example, Table 1 shows the binary assessment quantitative methods apply to variables associated with social vulnerability. However, transferring knowledge of variable correlations from historic disaster experience to alternative geographies, cultures and natural hazards leads to an *a priori* approach, and is not always appropriate. Considering the amount of media coverage and subsequent management of the Parrett catchment after the 2013/2014 flooding, it is surprising that this population is classified as less vulnerable than the England and Wales population. Using the 'Number of persons per hectare' indictor with vulnerability increasing with population density as in input is potentially results in an underestimate of social vulnerability in rural settings. Therefore, it is important to be mindful that the differences highlighted in the methodologies of this paper are just one aspect of the complexity involved in defining social vulnerability. To further investigate the influence the methodological approach has on the classification of social vulnerability, additional research is required which assess a range of different natural hazards, using a greater number of vulnerability indicators, and over a range of spatial scales.



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
