# Peer review of "A review of multivariate social vulnerability methodologies; a case study of the River Parrett catchment, Somerset"

_Natural Hazards and Earth System Sciences, 2016_

## Referee Comment (RC1) · Anonymous Referee #1 · 24 Feb 2016

Dear authors,

it was a great pleasure to review your paper. It's well written and includes very important information, not only for the academic research community but also for policy makers; in particular, for the national flood risk management authorities, such as EA/DEFRA and many more. The paper covers an important topic. However, I have four small comments, which I would like to express below:

1) First, I would like to ask you to include a broader discussion to the term social vulnerability in the first part of the paper. I would add a broader literature review on the term social vulnerability (roots, schools, limitation of each discussion), so you can frame both terms. Besides your literature I also would suggest following papers to add in your paper: Birkmann, J. and T. Welle. (2015): Assessing the risk of loss and damage: exposure, vulnerability and risk to climate-related hazards for different country classifications; International Journal of Global Warming; Birkmann, J.; Cardona, O.D; Carreno; L.; Barbat;, A.; Pelling, M.; Schneiderbauer, S.; Kienberger, S., Kelier, M.; Alexander, D.; Zeil, P. Welle, T. (2013): Framing vulnerability, risk and societal responses: the MOVE framework. Natural Hazards, 67(2): 193-211; Birkmann, Jörn (2006): Measuring Vulnerability to Promote Disaster-Resilient Societies: Conceptual Frameworks and Definitions, in: Birkmann, Jörn (Eds.): Measuring Vulnerability to Natural Hazards - Towards Disaster Resilient Societies, Tokyo, New York, Paris; United Nations University Press: 9-54; Fuchs, S.; Kuhlicke, C. & V. Meyer (2011): Vulnerability to natural hazards – the challenge of integration. Natural Hazards 58 (2). p. 609-619; Fuchs, S. (2009): Susceptibility versus resilience to mountain hazards in Austria – Paradigms of vulnerability revisited. Natural Hazards and Earth System Sciences 9 (2). p. 337-352; Hufschmidt (2011): A comparative analysis of several vulnerability concepts. Natural Hazards 58 (2). P. 621-643; Kuhlicke et al. (2011): A comparative analysis of several vulnerability concepts. Natural Hazards 58 (2). P. 789-810. Wisner et al. (2004) At risk: natural hazards, people's vulnerability and disasters.

2.) Please, provide a more critical reflection (in generally) on the use of social vulnerability indicators; what are the methodological limitations?

3) In your paper you selected a very special region. In particular, the community of Bridgwater shows a high socially problem within the country – please add some more paragraphs about the area (maybe this paper helps: Thaler & Levin-Keitel (2016): Multi-level stakeholder engagement in flood risk management – A question of roles and power: Lessons from England. In: Environmental Science & Policy. 55 (2): 292-301)

4) Last point: what is missing in the paper: the impact of your results/studies for the policy makers. What does your paper mean for the actual flood risk management policy in England? What does this mean for the actual localism/public-private debate in flood risk management in England (see Thaler

& Levin-Keitel (2016): Multi-level stakeholder engagement in flood risk management – A question of roles and power: Lessons from England. In: Environmental Science & Policy. 55 (2): 292-301 and Thaler & Priest (2014) Partnership funding in flood risk management: new localism debate and policy in England. Area 46 (4): 418-425). For the national authorities who are dealing with flood risk management also outside of the UK, such as US Army Corps.

---

## Short Comment (SC1) · 11 Mar 2016

A great many thanks for the review and feedback on the paper. Very helpful points and certainly things that myself and the co-author will look to address in the coming weeks.

I'm familiar with several of the papers and books listed in the first comment but am keen to read the others mentioned and see how I can integrate these into the paper.

Similarly, very good points on localism, the role of government and the wider implications of this research. We'll look to revise and re-issue in the coming weeks,

Many thanks,

Iain

---

## Author Comment (AC1) · 30 Mar 2016

**A review of multivariate social vulnerability methodologies; a case study of the River Parrett catchment, UK**

I. Willis[1] and J. Fitton[2]

[1]Birkbeck, University of London, United Kingdom
5      [2]University of Glasgow, Glasgow, United Kingdom

*Correspondence to:* I. Willis (iain.willis@jbarisk.com)

**Abstract.** In the field of disaster risk reduction (DRR), there exists a proliferation of research into different ways to measure, represent, and ultimately quantify a population's differential social vulnerability to natural hazards. Empirical decisions such as the choice of source data, variable selection, and weighting methodology can lead to large differences in the classification
10      and understanding of the 'at risk' population. This study demonstrates how three different quantitative methodologies (based on Cutter et al. (2003), Rygel et al. (2006), and Willis et al. (2010)) applied to the same England and Wales 2011 Census data variables in the geographical setting of the 2013/2014 floods of the Parrett river catchment, UK, lead to notable differences in vulnerability classification. Both the quantification of multivariate census data and resultant spatial patterns of vulnerability are shown to be highly sensitive to the weighting techniques employed in each method. The findings of such research highlight
15      the complexity of quantifying social vulnerability to natural hazards as well as the large uncertainty around communicating such findings to stakeholders in flood risk management and DRR practitioners.

**Introduction**

The impacts of a natural hazard event upon a population vary considerably depending upon the socioeconomic attributes of the people exposed to the hazard (O'Keefe et al., 1976; Yoon, 2012; Zakour and Gillespie, 2013). This concept can be termed
20      social vulnerability, however the exact definition of this term, and other associated concepts e.g. resilience, adaptive capacity, are contested within the literature (Brooks, 2003; Fuchs, 2009; Kuhlicke et al., 2011). These disparate views on social vulnerability are a consequence of models/frameworks to explain the relationship between hazard, risk, and vulnerability emanating from distinct schools of thought. Birkmann et al. (2013) lists these schools as including political ecology, social-ecology, vulnerability, disaster risk assessment, and climate change systems adaption. The definition of social vulnerability
25      from political ecology is used here, therefore it is defined as "the characteristics of a person or group and their situation that influences their capacity to anticipate, cope with, resist, and recover from the impact of a hazardous event" (Wisner et al. 2004, p. 11).

An individual's level of social vulnerability is multi-faceted, and determined by a number of spatially and temporally distant political, economic, and social 'root causes' (Birkmann et al., 2013; Watts and Bohle, 1993). These processes ultimately
30      manifest at a local scale into a range of 'unsafe conditions' e.g. living in dangerous locations, low income (see the Pressure and Release Model (PAR) developed by Wisner et al. (2004)). Natural hazards cannot be prevented, however the impact of natural hazards can be lowered by reducing the social vulnerability of the exposed population (Zakour and Gillespie, 2013). Therefore, there is great value in quantifying and spatially mapping 'unsafe conditions' i.e. a population's social vulnerability, to target mitigation and adaptation strategies at the areas that are both exposed and with high social vulnerability i.e. the most
35      at risk populations (Nelson et al., 2015; Rygel et al., 2006; Yoon, 2012). An often used method to quantify social vulnerability is based on the 'Hazards-of-Place' model (Cutter, 1996) which is a conceptual understanding of how 'unsafe conditions' interact at the local scale to produce a place vulnerability. Cutter et al. (2003) subsequently developed a quantitative methodology to identify and classify social vulnerability using census data, which became trademarked, known as the Social

Vulnerability Index (SoVI®). Whilst there are strengths and weaknesses of using such indicator and index based methodologies to assess social vulnerability, as detailed by Kuhlicke et al. (2011), the approach is used extensively e.g. Myers et al. (2008), Reid et al. (2009), Tapsell et al. (2002), Rygel et al. (2006), Willis et al. (2010), and Tomlinson et al (2011).

Despite a general consensus in social science about some of the main factors influencing an individual's social vulnerability e.g. age, income, health, education level (Adger et al., 2004; Cutter, 1996; Cutter et al., 2003; Wisner et al., 2004). However, there has been no agreement on a set of social vulnerability indicators for environmental hazards to use within an index (Cutter et al., 2003; Yoon, 2012). The data to include is constrained by the indicators relevance to the particular hazard(s) being assessed, and whether data are available and current (census data is often the primary data source). As a result, the number, and type of vulnerability indicators used within the construction of social vulnerability indices varies considerably depending on the type of analysis, and methods used (Nelson et al., 2015).

Once the relevant vulnerability indicators have been selected to construct an index, they are combined into a single metric. However, Yoon (2012, p. 824) states that "*there is still no consensus…on the quantitative methodology best suited to assess social vulnerability*". Within the literature, the predominant method used is a multivariate factorial method, in the form of principal component analysis (PCA) using census data e.g. Rygel et al. (2006), Boruff et al. (2005), Cutter et al. (2003), Clark et al. (1998). Willis et al. (2010) use another method which utilised a commercial geodemographic (Experian Mosaic Italy) classification as the main data source, and Gini coefficients to weight the vulnerability variables.

Yoon (2012) analysed the difference between a deductive and inductive approach when creating a vulnerability index, however there has been no further research into comparing different vulnerability methodologies. Therefore, there is limited information on whether all being equal, the different vulnerability methodologies classify the same people as highly vulnerable. The aim of this paper is to compare the social vulnerability indices produced when using three published methodologies: a method based on Cutter et al. (2003), a method using Pareto ranking based on Rygel et al. (2006), and a method with Gini coefficient weighting based on Willis et al. (2010). The area of the Parrett river catchment, UK, which was severely flooded in the winter of 2013/2014, will be used as a case study. If these approaches identify different populations as vulnerable, it raises a number of questions about how the 'at risk' population is defined. This paper will firstly, review the chosen vulnerability index methodologies, and describe the case study area. Secondly, the method used to compare the social vulnerability indices will be detailed. Finally, the results will be presented, and discussed.

**Quantitative approaches to measure social vulnerability**

Quantitative social vulnerability methodologies are predominantly based around the concept of indicators. That is to say, they are based on the a priori understanding that a given statistical variable, typically being socioeconomic or ethnographic, is highly correlated with an individual or group of people's inherent vulnerability before, during or after a given natural disaster. The qualitative research of such disaster experience includes historic evidence from various hurricanes, floods, earthquakes and famine (McMaster and Johnson 1987; Lew & Wetli 1996; Johnson and Ziegler 1986; Chakraborty et al. 2005; Dow and Cutter 2002; Burton et al. 1992; Morrow 1999; Dwyer et al. 2004). Such findings have subsequently guided the principles of quantitative researchers seeking to identify and model the most vulnerable population groups from the impact of future catastrophes. Aside from the indicator based approaches examined in this paper (Cutter et al 2003; Rygel et al 2006; and Willis et al 2010) it is important to note the influence of the wider global initiatives aimed at creating greater community resilience for disaster mitigation. The UN's Hyogo Framework (2005-2015) provided the contextual setting for much of this effort in the last 10 years and identified core aims focused on tools to help in disaster risk reduction (DDR), including *Priority Action 2,* specifically aimed to "*Identify, assess and monitor disaster risks and enhance early warning*" (UNISDR 2005) with specific reference to the use and application of vulnerability indicators. Though the concept of indicator based approaches has

historically been used to underpin economic theory (Hartmuth 1998; Reich and Stahmer 1983) or environmental indicators in the 1970's (Werner 1977; OECD 1993), the methodologies discussed in this research are aligned with the more recent sustainable development concept of indicators (Birkmann 2004).

Indicator based approaches can provide the practical means for practitioners in DRR to identify vulnerable population groups or communities to the risk(s) of a given peril. Similarly, these methodologies are not restricted in their spatial scale or scope, whether being a global 'hotspots' assessment of multiple natural hazard risk (Dilley et al 2005) or single peril, census based index examining flood vulnerability, as developed by Lindley et al (2011). It's important to be mindful that indicator approaches are not without their fundamental limitations. The "*definitions and drivers of vulnerability and indicators to measure them vary between industrialised and less-industrialised nations, especially where development pressures are inextricably linked to risk and vulnerability from local to global scales*" (Birkmann 2006, p. 304-305). Applying the concepts of social vulnerability, as evidenced by indicators in one contextual setting, does not translate that the same concepts can be applied or appropriate in another geography or spatial scale. Vulnerability is a dynamic notion, and thus, it is important to assess any indicator based approach within the political, environmental and socioeconomic landscape that it is being applied.

[revised manuscript text omitted]

apply the vulnerability scores within a pertinent context. By doing so, it was proposed that meaningful assessment could be

undertaken of the results within a realistic natural hazard setting. The Parrett catchment, in Dorset/Somerset, UK was chosen

as the case study area for this research (Figure 1). The Environment Agency (2009) report that the Parrett catchment is

approximately 1,700 km$^2$, and along with the River Parrett, includes the Isle, Tone, Yeo and Cary rivers which flow in a north

135    and westerly direction into an extensive lowland floodplain, before flowing out into the Bristol Channel via the Parrett Estuary.

The catchment contains approximately 300,000 people, however the catchment is predominately rural (only 4% is considered

urban), with three main urban centres (Yeovil, Taunton, and Bridgwater). The Environment Agency (2009) estimate that 3,300

properties are potentially exposed to a 1% annual probability flood event within the catchment, with this possibly rising to

over 6,600 properties in the future, due to the impacts of climate change. There is evidence that this rise is likely to occur as

140    the flooding in England and Wales in 2013/2014 is thought to be linked to human-induced climate change (Schaller et al.,

2016). Furthermore, Bridgwater was used as a case study area by Thaler & Levin-Keitel (2016) who identified the area as having a low capacity to engage in flood risk management due to the lack of socioeconomic structures i.e. cultural capital, income and interest.

[Figure]

145 **Figure 1: The location of the Parrett catchment, within the Somerset Levels area of south-west UK. The extent of the flooding in 2013/2014 is also shown.**

The UK experienced an unprecedented level of rainfall during the winter of 2013/2014, resulting in prolonged flooding in England and Wales, which is estimated to flooded 10,465 properties, and caused a total of £1.3 billion of economic damages (Chatterton et al., 2016). The rainfall flooded a 65 km$^2$ area of the Somerset Levels area of the River Parrett catchment

150 (Environment Agency 2015). Approximately 600 properties were flooded during this period, leaving a number of towns and villages cut off due to the high floodwaters. Flood waters persisted until March 2014 and the damage witnessed raised a national debate about the lack of dredging in the rivers throughout the Parrett catchment (Coghlan, 2014; Envionment Agency 2015;). This political pressure resulted in ministerial intervention and the subsequent production of 'The Somerset Levels and Moors Flood Action Plan', a 20-year scheme to mitigate future flood potential and increase the level of funding for flood

155 management in the region (Somerset County Council, 2014).

Alongside the physical damage of the Somerset Levels flooding, there has been limited consideration of the social vulnerability of those communities affected. Within the River Parrett catchment area there are a range of socioeconomic profiles, and while many of the most deprived communities (those located in urbanised areas such as Yeovil, Taunton and Bridgewater) were not adversely impacted, flood risk potential remains high. In the wider context of flood risk management, England and Knox

160 (2015, p, 7) show that in England *"levels of planned expenditure in flood risk management to 2021 do not appear to align with areas of significant flood disadvantage, or with wider deprivation",* i.e. the social vulnerability of the population potentially impacted by flooding currently has no bearing on spending decisions. In this instance, vulnerability to flooding used by England and Knox (2015) was derived using a method based on Cutter el al. (2003) by Lindley et al. (2011).

Given the prevalence of flood risk, range of socioeconomic characteristics, and combination of urban and rural populations

165 within the Parrett catchment, the area was seen as an ideal case study for this research. 
[revised manuscript text omitted]
 used by Willis et al. (2010) may lead to data outliers through the exponential loading of higher or lower vulnerability scores, though the concept of an inclusive methodology could arguably be more relevant than the selection bias of other approaches based on the PCA cardinality.

Whilst recognising the uncertainty that various statistical methods impose on indices, it is critical to note that the fundamental qualitative indicator based assumptions underlining social vulnerability concepts are arguably the greatest source of uncertainty. Transferring evidence of variable correlation from historic disaster experience to alternative geographies, cultures and natural hazards leads to an *a priori* approach with systemic uncertainty. Though qualitative evidence may be grounded in strong correlations between a statistical indicators (e.g. socioeconomic or ethnographic) and the polarisation of disaster experience during a given catastrophic event, there is inherent uncertainty as to whether such indicators can be successfully applied in a predictive model in another setting (whether temporal or spatial).

Despite the media coverage and subsequent management of the Parrett catchment after the 2013/2014 flooding, the OAC classifications and vulnerability indices presented here do not regard this population as being more vulnerable than the England and Wales average. Using the 'Number of persons per hectare' indictor with vulnerability increasing with population density results in an underestimate of social vulnerability in rural settings. Therefore, it is important to be mindful that the differences highlighted in the methodologies of this paper are just one aspect of the complexity involved in defining social vulnerability. To further investigate the influence the methodological approach has on the classification of social vulnerability, additional research is required which assess a range of different natural hazards, using a greater number of vulnerability indicators, and over a range of spatial scales.

The findings of this study also have implications in both how we convey the uncertainty of such vulnerability assessments as well as in the wider concern of UK flood defence management. Social vulnerability scores or metrics are typically provided as absolute values, but as this study has shown, there are numerous, equally plausible statistical methods that can lead to very different interpretations about the vulnerability of the same population. Similarly, in the wake of the December 2015 flooding in Yorkshire and Cumbria as well as the Somerset floods of 2013/2014, such research can help further inform local and national stakeholder debate in where UK flood defence funding is best focused to help the most disadvantaged. Likewise, social vulnerability indices focused on flood risk (Lindley et al 2011) can help advise the ongoing debate of localism and where government spending or private-public partnerships are best undertaken (Thaler and Priest 2014).

---

## Referee Comment (RC2) · Anonymous Referee #2 · 4 Apr 2016

Line 53 et seq.: It should be noted that principal components analysis is the most inductive method of all. It produces composite variables that in most cases have no inherent explanation. Rather than characterising any social process, it measures covariation whatever its cause may be.

Line 91: it's = it is

Line 132 et seq.: data were used...

Lines 135-7: How were these variables selected and why were the other 52 excluded?

Lines 155-166: Repetition vis-a-vis the previous page.

Line 276: Figure ?

Lines 276-8: Observations of this kind cry out for explanation. It seems that higher population density equals greater social vulnerability.

Figure 6: It is interesting how little the social vulnerability map corresponds with the flood map. I would have expected to see them overlaid.

This article is summed up by its own conclusions (lines 326-7), "the fundamental qualitative assumptions underlining [sic] social vulnerability are perhaps the first source of uncertainty in this process." The paper uses ill-justified variables and a highly inductive methodology (essentially a blind correlation exercise) to define a vague sort of 'social vulnerability' that seems to be independent of vulnerability to flooding, which is, in the first place, driven by flood hazard.
* * *

---

## Author Comment (AC2) · 5 Apr 2016

Thank you for the review of the paper. Please see author replies (AR) following each comment raised below;

Line 91: it's = it is – (AR) Agree, but suggest that neither are appropriate. Propose changing to 'its' to recognise the possessive form within this sentence.

Line 132 et seq: data were used... (AR) Agree. This could be changed to the plural form of 'data'

Lines 135-7: How were these variables selected and why were the other 52 excluded? (AR) I think this question is already directly addressed in the paragraph immediately

following and draws the reader to reference Tables 2 and 3 and how the variables selected are linked to previous studies from the literature showing how specific variables are correlated to social vulnerability (Lines 140-147)... "There were two main reasons for the seven initial indicators shown in Table 2. Firstly, as the focus of the study was to determine the difference that alternative weighting mechanisms may have on vulnerability scores, using fewer indicators made it easier to infer the influence of each methodology being reviewed. Secondly, not all census variables were eligible for inclusion in this study given that the focus was on determining factors that impact a neighbourhood's social vulnerability during extreme flooding. Whilst not exhaustive, Table 2 also provides example studies of where age, ethnicity, and disability have been shown to impact social vulnerability to support the selection of indicators within this study. Table 3 shows the correlation between the selected vulnerability indictors, with 'Persons aged 65 to 89' and 'Individuals day-to-day activities limited a lot or a little' (k005 and k035) showing the strongest relationship (0.687)." However, we will review the wording if variable selection is still not expressed clearly in this section.

Lines 155-166: Repetition vis-a-vis the previous page. (AR) Cannot see the relevance of this comment - there is no repetition here. Lines 152-163 discuss the data standardisation methodology and concept (i.e. why it's necessary to transform unformatted data and the Range standardisation method used). The previous page discusses initial 'variable selection', the rationale for this and the correlation of the variables. If the reviewer can kindly elaborate on what they mean?

Line 276: Figure ? (AR) An earlier figure was removed without editing the caption. This point was addressed in an earlier revision of the paper – please see the attachment to the previous reviewer's comments.

Lines 276-8: Observations of this kind cry out for explanation. It seems that higher population density equals greater social vulnerability. Figure 6: It is interesting how little the social vulnerability map corresponds with the flood map. I would have expected to see them overlaid. (AR) Agree, there is perhaps scope to elaborate on this further – for

the reviewer's reference, the population groups impacted by the Somerset levels largely consisted of small rural villages (many of whom were affluent farmers). This is not in itself a surprising correlation given that the region is by and large an extensive area of historic agricultural land (http://www.bbc.co.uk/news/uk-england-somerset-26080597). Whilst social vulnerability is shown to be more exacerbated in urban areas, there was no prior expectation that this would spatially correlate with flood risk.

This article is summed up by its own conclusions (lines 326-7), "the fundamental qualitative assumptions underlining [sic] social vulnerability are perhaps the first source of uncertainty in this process." The paper uses ill-justified variables and a highly inductive methodology (essentially a blind correlation exercise) to define a vague sort of 'social vulnerability' that seems to be independent of vulnerability to flooding, which is, in the first place, driven by flood hazard. (AR) Very strongly disagree with the reviewers' summary here. Oddly, it seeks to discredit/dismiss an entire body of literature (both qualitative and quantitative) on DRR that has evidenced how people's preparedness, response, mitigation, and recovery from a disaster are correlated with vulnerability traits linked to social indicators. I would recommend the reviewer read any of the following papers referenced for more on this topic; Wisner et al. 2004; McMaster and Johnson 1987; Lew  Wetli 1996; Johnson and Ziegler 1986; Chakraborty et al. 2005; Pulido 2000; Elliot and Pais 2006; Morrow 1999; Dwyer et al. 2004. The purpose of this study is to highlight/raise awareness about the uncertainties in quantitative methods - using just one case study area, a limited dataset, and a similar methodology, it seeks to raise the debate on uncertainty in quantifying social vulnerability more generally. The spatial correlation of flood risk is provided for context only and so that readers can see the implications such uncertainty has in a real setting.